# Prevalence, resistance profiles and factors associated with skin and soft-tissue infections at Jinja regional referral hospital: A retrospective study

**Fahad Lwigale**[1,2]*, **Daniel Kibombo**[1], **Simon Dembe Kasango**[1,3], **Dickson Tabajjwa**[1], **Collins Atuheire**[2], **Joseph Kungu**[2], **John Bosco Kalule**[2], **Morgan Otita**[1], **Francis Kakooza**[1], **Immaculate Nabukenya**[1,2], **Jonathan Mayito**[1], **Innocent B. Rwego**[2]

1 Global Health Security Program, Infectious Diseases Institute, College of Health Sciences, Makerere University, Kampala, Uganda, 2 School of Biosecurity, Biotechnical and Laboratory Sciences, College of Veterinary Medicine Animal Resources and Biosecurity, Makerere University, Kampala, Uganda, 3 Uganda National Health Research Organization, Ministry of Health, Kampala, Uganda

* fahadlwigale@gmail.com, flwigale@idi.co.ug

**Data Availability Statement:** All relevant data are available on Dryad (https://doi.org/10.5061/dryad. rjdfn2zkh).

## Abstract

Skin and soft-tissue infections (SSTI) are common cases of hospital-acquired infections with aetiological agents exhibiting antimicrobial resistance (AMR). This is a global public health predicament responsible for a high burden of infectious diseases and threatens the achievement of Sustainable Development Goals (SDGs), especially in Low- and Middle-Income countries (LMICs). This study determined the prevalence of SSTI, proportion of laboratory-investigated cases, AMR-profiles, and factors associated with SSTI and multi-drug resistance (MDR). This was based on records of patients suspected of SSTI for the period of 2019–2021 at Jinja Regional Referral Hospital. The analysis involved 268 randomly selected patient reports using WHONET 2022 and Stata 17 at the 95% confidence level. The prevalence of SSTI was 66.4%. Cases that involved laboratory testing were 14.1%. *Staphylococcus aureus* (n = 51) was the most isolated organism. MDR pathogens explained 47% of infections. Methicillin-resistant *Staphylococcus aureus* (MRSA) was up to 44%. In addition, 61% of Gram-negatives had the potential to produce extended-spectrum beta-lactamases (ESBL), while 27% were non-susceptible to carbapenems. Ward of admission was significantly associated with infection (aPR = 1.78, 95% CI: 1.00–3.18, p-value = 0.04). Age category (19–35) was an independent predictor for MDR infections (aPR = 2.30, 95% CI:1.02–5.23, p-value = 0.04). The prevalence of SSTI is high with MDR pathogens responsible for almost half of the infections. Gentamicin and ciprofloxacin can be considered for empirical management of strictly emergency SSTI cases suspected of *Staphylococcus aureus*. Given the high resistance observed, laboratory-based diagnosis should be increased to use the most appropriate treatment. Infection Prevention and Control (IPC) strategies should be heightened to reduce the prevalence of SSTI. Recognizing SSTI under the Global Antimicrobial resistance Surveillance System (GLASS) would lead to improved preparedness and response to AMR.

**Funding:** The author(s) received no specific funding for this work.

**Competing interests:** The authors have declared that no competing interests exist.

## Introduction

Antimicrobial resistance (AMR) is an emerging public health threat of concern globally [1, 2]. It has been noted to be responsible for negative social, economic, and health consequences; higher healthcare costs; increased Disability Adjusted Life Years (DALYs); and decreased economic growth [3, 4]. This burden is projected to increase in the near future if no proper attention is paid to managing it [3]. Improper use of antimicrobials is one of the main factors contributing to the development of AMR [1, 5, 6]. This has been reported in Uganda, which includes the prescription of antimicrobial agents for the wrong condition in lower health facilities [1, 5, 7]. Laboratory surveillance is conducted in Uganda as part of the mechanisms to tackle AMR in line with the global action plan for AMR [8]. This involves microbiology services such as culture and sensitivity (C&S) testing to enable identification of causative agents and the appropriate antimicrobial agents for managing individuals with infections, such as skin and soft-tissue infections (SSTI) [9, 10]. However, there is still a low coverage in this setting as most clinical case management is empirical.

Skin and soft-tissue infections are some of the most commonly encountered cases of hospital-acquired infections (HAI) and are characterized by AMR mainly among post-operative patients in low and middle-income countries (LMICs) [11–13]. These are a type of infection involving colonization and inflammation of the epidermis, dermis, and subcutaneous tissues [10, 14]. The colonizing agents such as bacteria are commonly from the hospital environment such as sinks, surgical beds, staff and wound dressings. These have been reported to be highly resistant to the commonly used antimicrobial agents [15–17]. This among other factors have been reported to influence the occurrence of SSTI [13, 18–21]. The prevalence of SSTI was between 10.3% to 15.6% in sub-Saharan Africa [22]. Positivity rates ranged from 81.9% to 92% in Uganda [23–25]. These wound infections were majorly due to Gram-negative organisms [6, 26, 27] and *Staphylococcus aureus* [2, 28–30]. A significant number of these bacteria are multi-drug resistant (MDR) [24, 26, 31] which can cause delayed healing.

There is insufficient utilization of microbiology laboratory services during infection management in some health facilities in Uganda [5, 23], leaving room for non-targeted therapy. Further, the Global Antimicrobial Resistance and antimicrobial use Surveillance System (GLASS) [8, 32] does not currently consider SSTI surveillance despite their high cultural yields and associated AMR observed in various microbiology laboratories [23, 24, 33]. This limits the availability of essential information such as SSTI trends, resistance rates, influencers and geographical distribution which is globally necessary for appropriate response against the AMR epidemic. In addition to inadequate research about microbiology service utilization to guide therapy, the level of SSTI and resistance patterns are unknown at Jinja Regional Referral Hospital, with little knowledge about factors influencing them. Lack of routine SSTI surveillance increases the risk for emergence and transfer of highly resistant pathogens to cause more acute, life-threatening events such as bloodstream infections and meningitis. This study therefore determined the prevalence of SSTI, proportion of SSTI that undergo laboratory investigation, common causative agents and their antibiotic resistance profiles, and factors associated with SSTI and MDR infections. This information would enable formulation or/and review of guidelines for better management of SSTI and improve regional antimicrobial stewardship practices for containment of AMR.

## Materials and methods

### Study design

This was a retrospective study based on the abstraction of socio-demographic and clinical information from charts of patients diagnosed with SSTI from January 2019 to December

2021. The data was accessed for analysis in June 2023. The study took place shortly after the establishment of microbiology services, majorly culture and sensitivity testing in 2018 and was made readily available for routine use in the region.

## Study setting

The study was carried out at Jinja RRH in the Eastern-central region of Uganda (**Fig 1**). The hospital is located in the center of Jinja city. Jinja is a focal point and refreshment area along the path from the Ugandan capital, Kampala towards the Kenyan border. This is a path that has encountered several traffic accidents in recent years, the majority of whose victims are managed at Jinja RRH [34–36]. This facility serves the Eastern-central region of Uganda which involves a population of approximately 4.5 million people from within Jinja and the surrounding areas such as Iganga, Mayuge, Bugiri, Kamuli, Buikwe, Lugazi, Kayunga, and Mukono districts among others [37]. The facility is equipped with a laboratory accredited by the South African National Accreditation System based on the requirements of ISO 15189 [38]. The laboratory results in this study were abstracted from the laboratory records. They were generated using the conventional microbiology methods for bacterial identification. Antimicrobial susceptibility testing(AST) was done using the Kirby-Bauer disk-diffusion method and interpreted according to the Clinical and Laboratory Standards Institute (CLSI) guidelines [39–41]. The laboratory observes internal quality control measures and engages in routine external quality assessments. Isolated organisms are periodically used for inter-laboratory comparison with the reference laboratory.

## Study population and sampling

The investigated population included records of patients who were managed for SSTI with or without laboratory testing. Records for both inpatients and outpatients were considered.

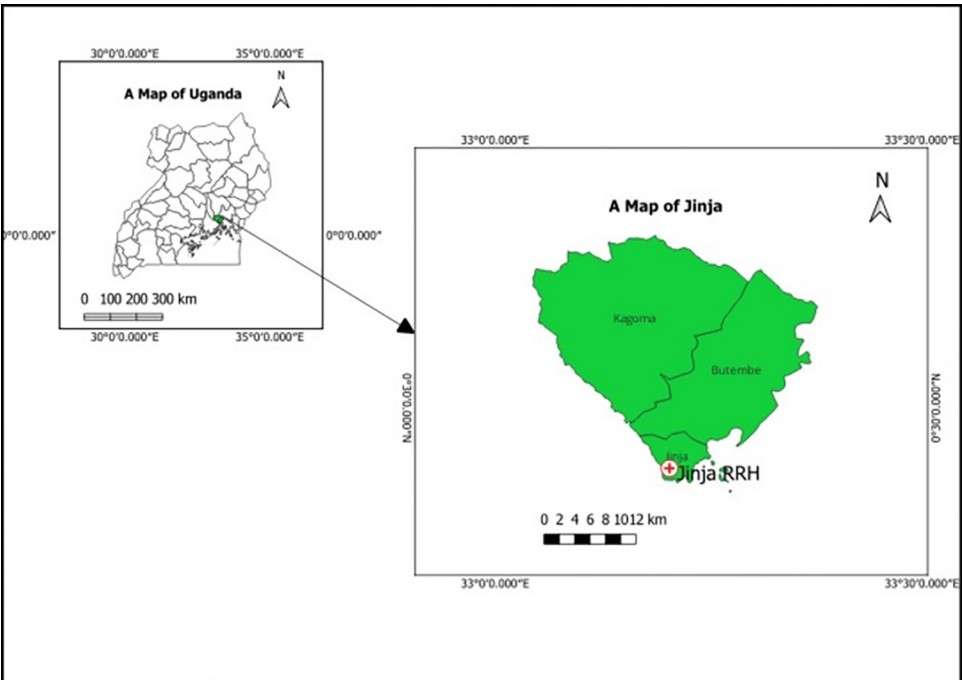

**Fig 1. A map showing the location of the study site.** This map was created using QGIS Desktop 3.32.1. The base layer is freely accessible from https://diva-gis.org/data. This can be shared under CC-BY license 4.0.

Laboratory records without updated results were not included. Only the first isolate of any patient was considered for analysis to generate antimicrobial resistance profiles.

Out of 526 laboratory patient records, a total of 268 reports were selected by systematic random sampling for the study. The sample size was calculated using the formula ($n = p(1-p)z^2/d^2$) where p = 20.8% [29], d = 5% and contingency for incomplete records = 5%. This had a power of 80% to estimate the prevalence and factors associated with SSTI at the 95% confidence level.

## Data collection

**Extraction of laboratory generated data.** The data sample frame with the necessary variables for the study was accessed on June, 30th 2023 and extracted from the electronic African Laboratory Information System (ALIS) of Jinja RRH laboratory into an Excel sheet. This included demographic and clinical data of patients who underwent laboratory testing including age, sex, ward, hospitalization history before testing, isolated organism, and AST results where applicable.

**Review of patients' files.** In addition to the existing laboratory-generated data, patients' files for the same study patients were sought and examined for more data necessary to investigate the associated factors. This was obtained using a predesigned data extraction tool transformed into the kobo-collect mobile application with kobo-toolbox open access software. The tool was piloted to confirm functionality and ability to obtain the required data before the actual study. This involved the entry of data for ten random patients admitted to the surgical ward into the electronic tool. This was saved on an online server and the aggregated data was downloadable in the form of a spreadsheet. Research assistants including a nurse, and records personnel were trained on the research tool in the same period and they became familiar with the data collection process. The collected data included admission periods, history of undergoing surgery, type of surgery, the theatre involved, and whether a patient was treated based on AST results from the laboratory.

**Data extraction from the District Health Information System (DHIS).** The overall number of patients diagnosed and treated for SSTI in the study period was obtained to aid the determination of the proportion of suspected SSTI that underwent microbiology testing for confirmation. This was done by examining standard Health Management Information System (HMIS 108 and HMIS 105:01) reports from the electronic DHIS2. The medical conditions considered for counting as part of SSTI included infections affecting the SSTI such as the middle ear, gangrenes, skin abscesses, and similar ones whether acute or chronic. These included the following as stated in the HMIS tools. Skin diseases (CD14), tetanus (CD15), otitis media (EN01), otitis externa (EN10), and burn injuries (OT04) from HMIS 105:01. Those in HMIS 108 included musculoskeletal and connective tissue diseases (LD04), cutaneous ulcers (LD09), osteomyelitis (CD11), tetanus (CD13), rheumatoid arthritis (RM01), septic arthritis (RM02), osteoarthritis (RM03), otitis media (EN01), injuries (IN01), diseases of the skin (LD03) and sepsis related to pregnancy (MC07). The total sum of diagnoses from the stated conditions was treated as the total number of patients with indication and treated for SSTI in the study period.

## Ethics statement

The Jinja Hospital Research and Ethics Committee (JREC) approved the study with registration number **JREC 395/2023.** The JREC granted a waiver for informed consent and medical records were anonymized prior to analysis for this study.

## Data analysis

The data collected was entered and cleaned in Microsoft Excel. Statistical analysis was performed using Stata 17. Categorical variables were summarized in the form of frequencies and percentages and presented by bar graphs and tables. Continuous variables were presented as means with standard deviation (SD). Prevalence of SSTI was calculated as overall and disaggregated prevalence. The overall prevalence was as a quotient of laboratory confirmed cases to the total number of suspected SSTI. Microbiology service utilization to confirm and manage suspected SSTI was estimated using two proportions; 1) Percentage of suspected SSTI investigated by C&S was obtained as a proportion of C&S tests done to the total sum of SSTI diagnoses (ΣC&S tests ÷ ΣSSTI Indications); 2) Percentage of patients managed based on microbiology (C&S) test results were calculated as the number of patients with de-escalation in treatment with antibiotics basing on C&S divided by the number of patients with positive C&S test (ΣPatients with de-escalation ÷ Σ Positive C&S tests). WHONET 2022 was used for antimicrobial susceptibility data analysis. Organisms with a minimum number of 30 isolates were considered individually to generate antimicrobial susceptibility profiles. Otherwise, organisms were grouped based on their microbiologic characteristics such as the order, Enterobacterales, and antibiograms generated for the group. Multi-drug resistance (MDR) was defined as an isolate resistant to at least three antibiotics of different clinical categories [42]. The independent variables were assessed for multicollinearity and had a Mean Variance Inflation Factor (VIF) of 1.58. Univariable analysis was performed to test for associations between the presence of SSTI and MDR etiology independently with possible predictors. Factors with a P-value <0.2 were followed up with multivariable analysis to fit a model using modified Poisson regression. Stepwise backward elimination was used and only significant variables were considered final predictors. Statistical significance was defined as a P-value of <0.05 at the 95% confidence level.

## Results

### Demographic characteristics

A total of 268 patient reports were included in the study. Of these, 55% (148/268) belonged to males. The patients had a mean age of 31 years (SD = 20.8) and most of the patients 31% (84/268) belonged to the age group of 19–35 years followed by 36–59 years 28% (76/268). Nearly a third of the patients 33% (88/268) were admitted to the surgical unit. Moreover, two-thirds of the study patients were undergoing antibiotic treatment before any microbiology testing was done (**Table 1**).

### Prevalence of skin and soft tissue infections

The prevalence of SSTI was 66.4% (95% CI = 60.70–72.10) based on the 178 laboratory-confirmed positive cases. Among these cases, 56.7% (101/178) were males. The most affected age groups were those between 19 and 35 years 29.8% (53/178) followed by 36–59 years, 27.5% (49/178). Polymicrobial growth was observed in 6.7% (18/268) of the cases. Of these, *Candida species*, 2.7% (5/18), were the major co-infection.

The pediatric ward had the highest prevalence of SSTI of up to 80% (8/10) while the accidents and emergency unit had the lowest, 43.7% (7/ 16) (**Table 1**).

Approximately 3,720 cases were diagnosed and treated for SSTI during the study period. Only 14.1% (526/3720) of C&S tests were done among patients suspected of SSTI. In 2019, only 8.1% (104/1278) of suspected SSTI cases underwent laboratory confirmation while 21.1%

**Table 1. Demographic characteristics of the study population with category SSTI prevalence.**

| Variable | | Number (n) | Percentage (%) | Number with SSTI (n) | Prevalence of SSTI (%) |
|---|---|---|---|---|---|
| Year of Case | 2019 | 44 | 16.4 | 31 | 70.5 |
| | 2020 | 120 | 44.8 | 75 | 62.5 |
| | 2021 | 104 | 38.8 | 72 | 69.2 |
| Sex | Male | 148 | 55.2 | 101 | 68.2 |
| | Female | 120 | 44.8 | 77 | 64.2 |
| Hospital admission >48hrs | Yes | 119 | 44.4 | 78 | 65.5 |
| | No | 98 | 36.5 | 61 | 62.2 |
| | Unknown | 51 | 19.0 | 39 | 76.5 |
| Age Category | ≤12 | 48 | 17.9 | 34 | 70.8 |
| | 13–18 | 31 | 11.6 | 21 | 67.7 |
| | 19–35 | 84 | 31.3 | 53 | 63.1 |
| | 36–59 | 76 | 28.4 | 49 | 64.5 |
| | ≥60 | 29 | 10.8 | 21 | 72.4 |
| Ward/Department | Accidents and emergency | 16 | 6.0 | 7 | 43.8 |
| | Gynecology | 13 | 4.9 | 7 | 53.8 |
| | Maternity | 15 | 5.6 | 8 | 53.3 |
| | Medical | 15 | 5.6 | 8 | 53.3 |
| | Surgical | 88 | 33 | 66 | 75.0 |
| | Outpatient Department | 38 | 14.2 | 25 | 65.8 |
| | Orthopedics | 34 | 12.7 | 20 | 58.8 |
| | Private wing | 8 | 3.0 | 6 | 75.0 |
| | Pediatrics | 10 | 3.7 | 8 | 80.0 |
| | Others | 31 | 11.6 | 23 | 74.2 |
| On antibiotics before testing | Yes | 176 | 65.7 | 117 | 66.5 |
| | No | 92 | 34.3 | 61 | 66.3 |
| Undergoing surgery | Yes | 11 | 4.1 | 6 | 54.5 |
| | No | 98 | 36.6 | 65 | 66.3 |
| | Unknown | 159 | 59.3 | 107 | 67.3 |

Prevalence per category was derived from number of laboratory-confirmed cases among clinically suspected SSTI. This was disaggregated according to the study population characteristics including admission history and other patient clinical features. n -Number of study cases

(226/1072) and 14.3% (196/1370) were laboratory confirmed in 2020 and 2021 respectively (**Fig 2**).

Only 4.1% (11/268) of the patients' records were found to have C&S test results in patient files. The treatment records for outpatients were not readily available. Therefore, the representative percentage of de-escalation based on the C&S report could not be estimated.

## Antimicrobial resistance patterns of selected bacteria responsible for skin and soft-tissue infections

**Major bacteria responsible for the observed SSTI.** There were 203 organisms isolated from the clinical samples. Of these isolates, 58.6% (119/203) were Gram-negative bacteria, 25.6% (52/203) were Gram-positive cocci and 3.4% (7/203) were yeasts. The majority were *Staphylococcus aureus*, 25.1% (51/203); followed by *Escherichia coli*, 13.7% (28/203); *Klebsiella species*, 8.9% (18/203); *Citrobacter species*, 8.4% (17/203); and *Proteus species*, 4.4% (9/203). Non-*Enterobacterales* were mainly made up of *Pseudomonas species* 5.4% (11/203) and

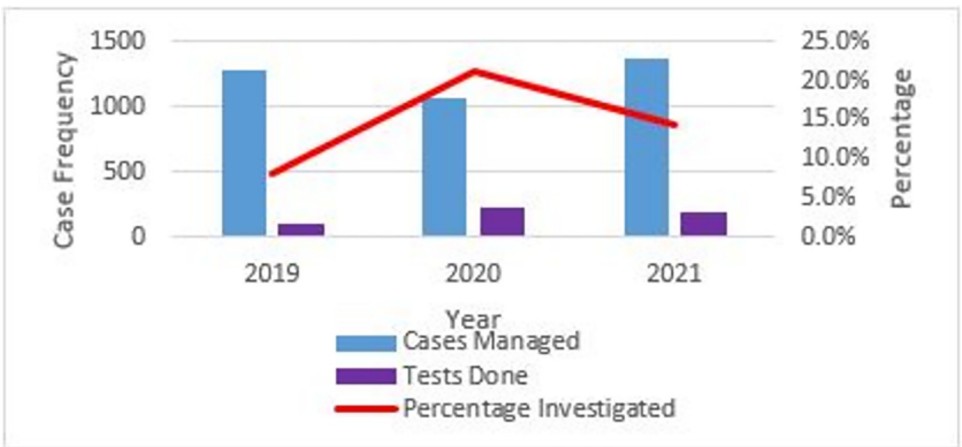

**Fig 2. Proportion of cases that underwent laboratory investigation.** Less than 30% of suspected SSTI cases involved laboratory testing in all the year periods studied.

*Acinetobacter species* 3.4% (7/203). Up to 3.4% (7/203) of isolates from the SSTI were *Candida species*. Other bacterial isolates also included *Coagulase Negative Staphylococci* (*CoNS*).

**Percentage resistance of the bacteria to common antibiotics.** Close to 47% (79/171) of the isolated bacteria were MDR pathogens. Among the Gram-negative bacteria, 61.3% (73/119) were resistant to third-generation cephalosporins and hence possible ESBL producers while 27.7% (33/119) were non-susceptible to carbapenems. All the tested isolates for *Staphylococcus aureus* were resistant to penicillin G 100% (23/23) (**Fig 3**). Over 44.4% (8/18, 95% C.I: 22.40–68.70) of the tested isolates were methicillin-resistant *Staphylococcus aureus* (MRSA; **Table 2**).

The highest percentage resistance among *Enterobacterales* altogether was against ampicillin 97.1% (34/35). This group was least resistant to meropenem 0% (0/5) and imipenem 15.6% (7/45) (**Table 3**).

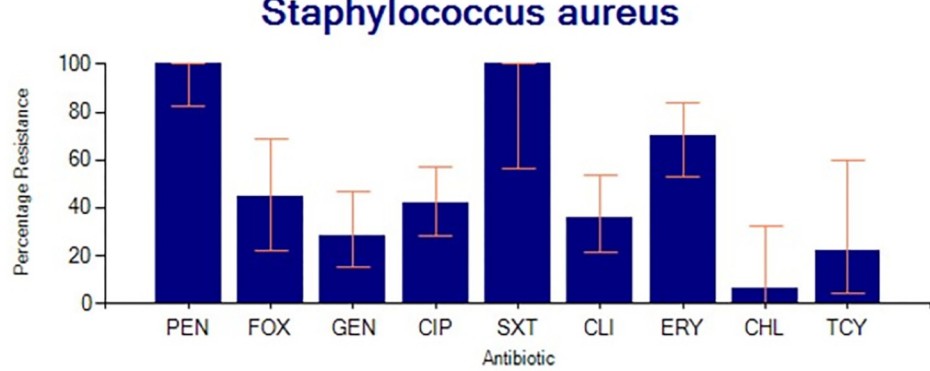

PEN-Penicillin G; FOX-Cefoxitin; GEN-Gentamicin; CIP-Ciprofloxacin; SXT-Co-trimoxazole; CLI-Clindamycin; ERY-Erythromycin; CHL-Chloramphenicol; TCY-Tetracycline

**Fig 3. Percentage resistance for *Staphylococcus aureus*.** The highest level of resistance was observed to be against penicillin G.

**Table 2. Antimicrobial susceptibility profile for *Staphylococcus aureus*.**

| Antibiotic name | Antibiotic class | Concentration(µg) | Breakpoints | Number Tested | %R | %I | %S | %R, 95%C.I. | %S, 95%C.I. |
|---|---|---|---|---|---|---|---|---|---|
| *Cefoxitin | Cephems | 30 | S > = 22 | 18 | 44.4 | 0.0 | 55.6 | 22.40–68.70 | 31.30–77.60 |
| Chloramphenicol | Phenicols | 30 | 13–17 | 16 | 6.3 | 12.5 | 81.3 | 0.30–32.30 | 53.70–95.00 |
| Ciprofloxacin | Quinolones | 5 | 16–20 | 48 | 41.7 | 16.7 | 41.7 | 27.90–56.70 | 27.90–56.70 |
| Clindamycin | Lincosamides | 2 | 15–20 | 36 | 36.1 | 16.7 | 47.2 | 21.30–53.80 | 30.80–64.30 |
| Erythromycin | Macrolides | 15 | 14–22 | 37 | 70.3 | 21.6 | 8.1 | 52.80–83.60 | 2.10–23.00 |
| Gentamicin | Aminoglycosides | 10 | 13–14 | 35 | 28.6 | 8.6 | 62.9 | 15.20–46.50 | 44.90–78.00 |
| Penicillin G | Penicillins | 10 units | S > = 29 | 23 | 100.0 | 0.0 | 0.0 | 82.20–100.00 | 0.00–17.80 |
| Tetracycline | Tetracyclines | 30 | 15–18 | 9 | 22.2 | 44.4 | 33.3 | 3.90–59.80 | 9.00–69.10 |
| Trimethoprim/Sulfamethoxazole | Folate pathway inhibitors | 1.25/23.75 | 11–15 | 7 | 100.0 | 0.0 | 0.0 | 56.10–100.00 | 0.00–43.90 |

Antimicrobial Susceptibility for *Staphylococcus aureus* as determined by Kirby-Bauer method interpreted as R-Resistant; S-Susceptible; and I-Intermediate. C. I-Confidence Interval for percentage Resistance at 95% level.

*Note: Cefoxitin is the recommended surrogate test agent for determining the susceptibility of *Staphylococcus aureus* to Oxacillin or Methicillin using the disk-diffusion method [39–41]. *Staphylococcus aureus* isolates that are resistant to Cefoxitin are regarded as Methicillin Resistant *Staphylococcus aureus* (MRSA).

Non-*Enterobacterales* composed of *Pseudomonas* and *Acinetobacter* species, together (n = 18) had a percentage resistance of 55.6% (7/12) for piperacillin, 33.3% (1/3) for amikacin, 50% (7/14) for ceftazidime, 29.4% (5/17) for ciprofloxacin, 18.2% (2/11) for gentamicin, and 14.3% (2/14) for imipenem (**Fig 4**).

**Factors associated with SSTI.** Patients in the surgical ward were significantly more likely to develop an SSTI compared to those in the accident and emergency (A&E) ward (aPR = 1.78, 95%, CI:1.00–3.18, p = 0.04). Age, gender, hospital admission hours, and status of antibiotic use before testing were not independently associated with risk for SSTI (**Table 4**).

**Factors associated with MDR pathogens responsible for SSTI.** Age was the only factor significantly associated with MDR, where patients aged 19 to 59 years were over two times more likely to have MDR pathogens compared to those who were 12 years or younger (aPR = 2.30, 95%CI:1.02–5.23, p = 0.04) (**Table 5**).

**Table 3. Antimicrobial susceptibility profile for *Enterobacterales*.**

| Antibiotic name | Antibiotic class | Concentration(µg) | Breakpoints | Number Tested | %R | %I | %S | %R, 95%C.I. | %S,95%C.I. |
|---|---|---|---|---|---|---|---|---|---|
| Amikacin | Aminoglycosides | 30 | 15–16 | 19 | 15.8 | 26.3 | 57.9 | 4.20–40.50 | 34.00–78.90 |
| Amoxicillin/Clavulanic acid | Beta-lactam Inhibitors | 20/10 | 14–17 | 7 | 71.4 | 0.0 | 28.6 | 30.30–94.90 | 5.10–69.70 |
| Ampicillin | Penicillins | 10 | 14–16 | 35 | 97.1 | 2.9 | 0.0 | 83.40–99.90 | 0.00–12.30 |
| Cefotaxime | Cephalosporin III | 30 | 23–25 | 18 | 77.8 | 16.7 | 5.6 | 51.90–92.60 | 0.30–29.40 |
| Ceftazidime | Cephalosporin III | 30 | 18–20 | 38 | 73.7 | 10.5 | 15.8 | 56.60–86.00 | 6.60–31.90 |
| Cefuroxime | Cephalosporin II | 30 | 15–17 | 10 | 80.0 | 0.0 | 20.0 | 44.20–96.50 | 3.50–55.80 |
| Chloramphenicol | Phenicols | 30 | 13–17 | 78 | 46.2 | 10.3 | 43.6 | 34.90–57.80 | 32.60–55.30 |
| Ciprofloxacin | Fluoroquinolone | 5 | 22–25 | 83 | 51.8 | 9.6 | 38.6 | 40.60–62.80 | 28.30–49.90 |
| Gentamicin | Aminoglycosides | 10 | 13–14 | 71 | 33.8 | 12.7 | 53.5 | 23.30–46.10 | 41.40–65.30 |
| Imipenem | Carbapenems | 10 | 20–22 | 45 | 15.6 | 8.9 | 75.6 | 7.00–30.10 | 60.10–86.60 |
| Meropenem | Carbapenems | 10 | 20–22 | 5 | 0.0 | 0.0 | 100.0 | 0.00–53.70 | 46.30–100.0 |
| Tetracycline | Tetracyclines | 30 | 12–14 | 18 | 72.2 | 0.0 | 27.8 | 46.40–89.30 | 10.70–53.60 |
| Trimethoprim/Sulfamethoxazole | Folate pathway inhibitors | 1.25/23.75 | 11–15 | 21 | 90.5 | 4.8 | 4.8 | 68.20–98.30 | 0.20–25.90 |

Antimicrobial Susceptibility for the major Gram-negative organisms as determined by Kirby-Bauer method interpreted as R-Resistant; S-Susceptible; and I-Intermediate. C.I-Confidence Interval for percentage Resistance at 95% level.

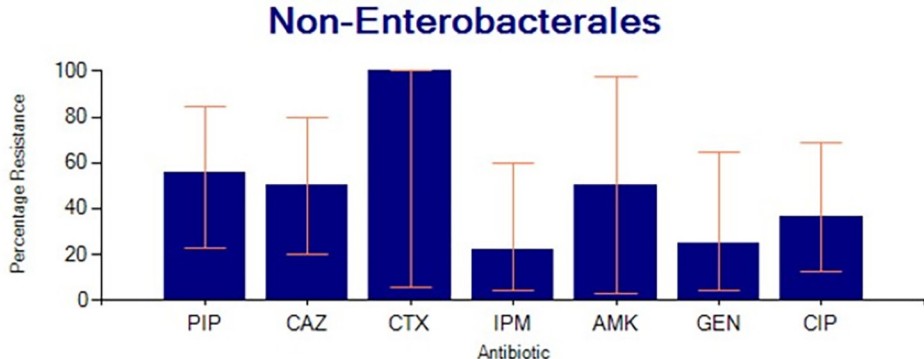

PIP-Piperacillin; CAZ-Ceftazidime; CTX-Cefotaxime; IPM-Imipenem; AMK-Amikacin; GEN-Gentamicin; CIP-Ciprofloxacin

**Fig 4. Percentage resistance for Non-*Enterobacterales*.** This group involved Pseudomonas species and Acinetobacter species combined for analysis.

## Discussion

This study provides the most recent epidemiology of SSTI and their resistance profiles in a Ugandan tertiary healthcare facility. The prevalence of SSTI was 66.4%, which was comparable to the prevalence reported in Pakistan (68.5%) [6], Sierra Leone (62.1%) [26] and Ethiopia (70%) [33]. However, studies in the same settings including Mbarara RRH (81.9%, 92%) [23, 24] and Mulago National Referral hospital (85%) [25] observed higher levels of infection. Much lower levels of skin infections, 1.5% [43], 3.1–4.4% [44], and 10.3–15.6% [22], have also been reported in China, Uganda and SSA respectively. These differences could have resulted from variations in sensitivity and specificity of the methods used to diagnose the infections. The clinical diagnostic approach based on physical examination was applied in some studies [43, 44] compared to the laboratory detection by culture and sensitivity used in this research and previous similar studies [23, 25]. Differences in observing infection prevention and control, immune status and climate change in different geographical regions are among the other possible causes of the variable prevalence.

Positive cases in which more than one aetiological agent was isolated (Polymicrobial growth) were 6.7%. This has been observed in other studies [12, 28, 31]. In this study, *Candida* species were involved in more polymicrobial infections compared to the cases solely by a fungus, indicating that an SSTI by a fungal organism is more likely to occur with an existing bacterial agent. This could have been due to the commensal relationship between the two organism types. A higher proportion of mixed infections (21.4%) has previously been observed compared to only fungal infections (5.8%) [45]. Also, there are significant interactions between bacteria and fungi to form biofilms reported to complicate healing, especially in chronic wound infections [46]. This calls for further utilization of appropriate diagnostics to detect the fungal infections to limit unnecessary and prolonged use of antibiotics especially among patients with chronic deep tissue infections [46].

In this study, sex was not associated with SSTI or MDR as similarly observed in Benin [18] but contrary to other studies where sex was significantly associated with a higher risk for infection [19, 26]. The age groups most affected by SSTI were those between 19 and 35 years followed by those between 36 and 59 years with SSTI in the 19–59 group being more likely to be due to MDR pathogens compared to those less than 12 years. This contrasts previous studies that reported a higher likelihood of infection among those above 35 years [47, 48]. This might

**Table 4. Factors associated with skin and soft tissue infections.**

| Variable | Infection | | Univariable Analysis | Multivariable Analysis |
|---|---|---|---|---|
| Ward, n (%) | No (n = 90) | Yes (n = 178) | cPR(95%CI), p-value | aPR(95%CI), p-value |
| A&E | 9(10.0) | 7(3.9) | 1.00 | 1.00 |
| Gynecology | 6(6.7) | 7(3.9) | 1.23(0.58–2.61) 0.58 | 1.32 (0.61–2.88) 0.48 |
| Maternity | 7(7.8) | 8(4.5) | 1.22(0.59–2.53)0.59 | 1.30 (0.61–2.77) 0.48 |
| Medical | 7(7.8) | 8(4.5) | 1.22(0.59–2.53) 0.59 | 1.28 (0.61–2.70) 0.51 |
| OPD | 13(14.4) | 25(14.0) | 1.50(0.82–2.75) 0.18 | 1.60 (0.86–2.98) 0.13 |
| Orthopedics | 14(15.6) | 20(11.2) | 1.34(0.72–2.51) 0.35 | 1.43 (0.76–2.69) 0.26 |
| Other | 8(8.9) | 23(12.9) | 1.69 (0.94–3.07) 0.08 | 1.81 (0.98–3.35) 0.05 |
| Private | 2(2.2) | 6(3.4) | 1.71(0.86–3.40) 0.12 | 1.86 (0.91–3.77) 0.08 |
| Pediatrics | 2(2.2) | 8(4.5) | 1.83(0.97–3.46) 0.06 | 1.78 (0.92–3.46) 0.08 |
| Surgical | 22(24.4) | 66(37.1) | 1.71(0.97–3.03) 0.06 | 1.78 (1.00–3.18) **0.04** |
| Sex, n (%) | | | | |
| Female | 43(47.8) | 77(43.3) | 1.00 | 1.00 |
| Male | 47(52.2) | 101(56.7) | 1.06(0.89–1.26) 0.48 | 1.04 (0.86–1.24) 0.70 |
| Age Group, n (%) | | | | |
| 12&below yrs. | 14(15.6) | 34(19.1) | 1.00 | 1.00 |
| 13-18yrs | 10(11.2) | 21(11.8) | 0.95(0.70–1.29) 0.77 | 0.97 (0.70–1.33) 0.84 |
| 19-59yrs | 31(34.4) | 53(29.8) | 0.89(0.69–1.13) 0.35 | 0.90 (0.68–1.19) 0.47 |
| 60+yrs | 35(38.9) | 70(39.3) | 0.94(0.75–1.18) 0.60 | 0.92 (0.71–1.19) 0.53 |
| Undergoing surgery, n (%) | | | | |
| No | 85(94.4) | 172(96.6) | 1.00 | |
| Yes | 5(5.6) | 6(3.6) | 0.81(0.47–1.40) 0.46 | |
| Surgical type, n (%) | | | | |
| Elective | 2(2.2) | 3(1.7) | 1.00 | 1.00 |
| Others | 88 (97.7) | 175 (98.3) | 1.10 (0.54–2.28) 0.77 | 0.52 (0.20–1.30) 0.16 |
| Hospital admission >48hrs, n (%) | | | | |
| No | 49(54.4) | 100 (56.2) | 1.00 | 1.00 |
| Yes | 41(45.6) | 78(43.8) | 0.97(0.82–1.16) 0.78 | 1.004 (0.79–1.27) 0.97 |
| Year of case | | | | |
| 2019 | 13 (14.4) | 31 (17.4) | 1.00 | 1.00 |
| 2020 | 45 (50) | 75 (42.1) | 0.89 (0.70–1.12) 0.32 | 0.96 (0.73–1.27) 0.79 |
| 2021 | 32 (35.6) | 72 (40.5) | 0.98 (0.78–1.23) 0.88 | 1.05 (0.80–1.37) 0.73 |
| Antibiotic use before testing | | | | |
| No | 31 (34.4) | 61 (34.3) | 1.00 | 1.00 |
| Yes | 59 (65.6) | 117 (65.7) | 1.002 (0.84–1.20) 0.97 | 1.04 (0.82–1.32) 0.72 |
| Type of theatre | | | | |
| Others | 86 (95.6) | 172 (96.6) | 1.00 | |
| Main theatre | 4 (4.4) | 6 (3.4) | 1.11 (0.66–1.86) 0.68 | |

Table shows outcomes from the Univariable and Multivariable regression analysis for factors associated with SSTI. cPR–Crude Prevalence Ratio, aPR–Adjusted Prevalence Ratio, p-value–Probability value, CI–Confidence Interval

be because this age group is the most active in life, prone to injuries and therefore exposure to antibiotics during treatment of the injuries, increasing their risk for MDR infections. Additionally, these are more likely to access antibiotics through self-medication, which increases the risk of AMR. Age has previously not been associated with MDR SSTI elsewhere [49].

The proportion of patients tested in the microbiology laboratory for the management of SSTI was found to be 14.1%, which was slightly less than the 23% observed in California, USA

**Table 5. Factors associated with multi-drug resistance (MDR) pathogens among patients with skin and soft tissue infections.**

| Variable | MDR | | Univariable Analysis | Multivariable Analysis |
|---|---|---|---|---|
| Ward, n (%) | No (n = 189) | Yes (n = 79) | cPR(95%CI), p-value | aPR(95%CI), p-value |
| A&E | 12(6.4) | 4(5.1) | 1.00 | 1.00 |
| gynecology | 8(4.2) | 5(6.3) | 1.54(0.51–4.60) 0.44 | 1.21 (0.39–3.74) 0.74 |
| maternity | 11(5.8) | 4(5.1) | 1.07(0.32–3.53)0.91 | 0.88 (0.25–3.09) 0.84 |
| medical | 12(6.4) | 3(3.8) | 0.80(0.21–3.00) 0.74 | 0.71 (0.19–2.67) 0.61 |
| OPD | 28(14.8) | 10(12.7) | 1.05(0.39–2.87)0.92 | 1.05 (0.39–2.80) 0.91 |
| Orthopedics | 25(13.2) | 9(11.4) | 1.06(0.38–2.93) 0.91 | 0.84 (0.29–2.41) 0.73 |
| Other | 22(11.6) | 9(11.4) | 1.16(0.42–3.20) 0.77 | 1.50 (0.54–4.15) 0.43 |
| Private | 5(2.7) | 3(3.8) | 1.50(0.44–5.16) 0.52 | 1.49 (0.45–4.95) 0.51 |
| Pediatrics | 8(4.2) | 2(2.5) | 0.80(0.18–3.60) 0.77 | 1.49 (0.29–7.39) 0.62 |
| Surgical | 58(30.7) | 30(38.0) | 1.36 (0.56–3.35) 0.49 | 1.18 (0.47–2.94) 0.71 |
| Sex, n (%) | | | | |
| Female | 88(46.6) | 32(40.5) | 1.00 | 1.00 |
| Male | 101(53.4) | 47(59.5) | 1.19(0.81–1.74) 0.36 | 1.25 (0.84–1.88) 0.27 |
| Age, n (%) | | | | |
| 12&below yrs. | 40(21.2) | 8(10.1) | 1.00 | 1.00 |
| 13-18yrs | 22(11.6) | 9(11.4) | 1.74(0.75–4.03) 0.19 | 2.14 (0.79–5.73) 0.13 |
| 19-59yrs | 54(28.6) | 30(38.0) | 2.14 (1.06–4.29) 0.03 | 2.30 (1.02–5.23) **0.04** |
| 60+yrs | 73(38.6) | 32(40.5) | 1.82(0.91–3.67) 0.09 | 1.96 (0.87–4.12) 0.10 |
| Undergoing surgery, n (%) | | | | |
| No | 181(95.8) | 76(96.2) | 1.00 | |
| Yes | 8(4.2) | 3(3.8) | 0.92(0.34–2.46) 0.87 | |
| Surgical type, n (%) | | | | |
| Elective | 2(1.1) | 3(3.8) | 3.69 (0.60–22.52) 0.15 | |
| Others | 187 (98.9) | 76(96.2) | 1.00 | |
| Hospital admission >48hrs, n (%) | | | | |
| No | 112(59.3) | 37(46.8) | 1.00 | 1.00 |
| Yes | 77(40.8) | 42(53.2) | 1.42(0.98–2.05) 0.06 | 1.17 (0.71–1.94) 0.54 |
| Year of case | | | | |
| 2019 | 36 (19) | 8 (10.1) | 1.00 | 1.00 |
| 2020 | 85 (45) | 35 (44.3) | 1.60 (0.81–3.19) 0.17 | 1.34 (0.61–3.03) 0.45 |
| 2021 | 68 (36) | 36 (45.6) | 1.90 (0.96–3.76) 0.06 | 1.63 (0.75–3.55) 0.21 |
| Antibiotic use before testing | | | | |
| No | 72 (38.1) | 20 (25.3) | 1.00 | 1.00 |
| Yes | 117 (61.9) | 59 (74.7) | 1.54 (0.99–2.39) 0.05 | 1.28 (0.73–2.24) 0.38 |
| Type of theatre | | | | |
| Others | 182 (96.3) | 76 (96.2) | 1.00 | 1.00 |
| Main theatre | 7 (3.7) | 3 (3.8) | 1.02 (0.39–2.68) 0.97 | 0.91 (0.34–2.46) 0.85 |

Table shows outcomes from the univariable and multivariable regression analysis for factors associated with multi-drug resistance. cPR–Crude Prevalence Ratio, aPR–Adjusted Prevalence Ratio, p-value–Probability value, CI–Confidence Interval

[50]. Treatment records for outpatients were not available while only 4.1% (11/268) of the study patients' records were found to have culture and sensitivity result reports in inpatient files. These low numbers could be due to the non-electronic system used for patients' records and poor communication between attending clinicians and the laboratory. This increases chances for misdiagnosis and poor choice of antibiotics to manage cases.

Most of the isolated bacteria were Gram-negative as similarly observed by other studies in which the Gram-negative accounted for most of the infections ranging from 72.9–91% [19, 26, 27]. However, *Staphylococcus aureus* was individually responsible for most of the infections observed in the current study. This is similar to studies conducted elsewhere [14, 28, 29]. *Staphylococcus aureus* was followed by *Escherichia coli* then other bacteria such as *Klebsiella* species, *Citrobacter* species, and *Pseudomonas* species. Similar organism rates have been reported in China, Ethiopia, and Rwanda [12, 27, 43].

This study observed that close to 47% of the infections were due to MDR pathogens, which is greater than the 22.6% prevalence observed in Poland [30]. This difference could be due to the employment of better infection prevention measures compared to the Ugandan settings. Other studies also reveal that a large number of the bacteria responsible for the SSTI are MDR [24, 26, 31].

Up to 61% of Gram-negative bacteria were resistant to third-generation cephalosporins hence the possible presence of ESBL producers, which was similar to 59.2% in Sierra Leone [26]. Meanwhile, 27% were non-susceptible to carbapenems which is higher than the 8.2% observed among the Enterobacterales [26]. The study observed MRSA levels of 44% which is comparable to Ethiopia (49%) [33] but less than levels reported in Saudi Arabia(65.4%) [51]. *This is greater than what was reported in previous studies from* Poland *(*23.6%) [30]. The observed resistance to common antibiotics could be due to the increased use of broad-spectrum antibiotics especially ceftriaxone for routine suspected infections in the local community.

All the *Staphylococcus aureus* isolates tested were resistant to penicillin G. However, gentamicin and ciprofloxacin had higher sensitivity compared to the other agents as similarly observed in Ethiopia [33]. *Enterobacterales* on the other hand showed the highest resistance to ampicillin. A similar observation was reported earlier [2]. The current study shows that the best agents for managing infections due to *Enterobacterales* currently include imipenem, gentamicin, and chloramphenicol. Isolates of *Acinetobacter* species and *Pseudomonas* species together (n = 18) had a percentage resistance of 55.6%, 50%, 50%, 36.4%, 25%, and 22.2% against piperacillin, amikacin, ceftazidime, ciprofloxacin, gentamicin, and imipenem respectively. However, their number was less than the threshold necessary to generate reliable antibiograms. Therefore, recommendations about their empirical management cannot be appropriately made based on the available information. Previously, piperacillin plus tazobactam has been recommended for use against *Pseudomonas aeruginosa* [6].

Patients in the surgical ward were 1.8 times more likely to develop an infection compared to those in the A&E ward. The outpatient department also had a lower prevalence of infection compared to the surgical ward. This could be due to the possession of open wounds from surgical repair that increase their liability to acquiring infections. Other studies have significantly associated infection with a history of surgery and the admitting unit [12, 18].

Patients with prior antibiotic exposure before microbiology testing were not more likely to have an SSTI due to MDR aetiologic agent compared to those who were unexposed. A similar outcome was observed earlier [31]. The factors that had positive associations with MDR infections included type of ward, type of theatre, gender of the patient, and year of case. However, these were not statistically significant.

This study had some limitations, including being based at a single facility and missing data for some variables due to the retrospective design. Out-patients had no treatment files available and some inpatient files could not be located due to the manual hardcopy filing system. This limited the ability to obtain information such as surgical history, theatre involved, length of admission, and treatment records. These were recorded as unknown for some patients. Individual variables with insignificant data (Less than 30 observations) could not be concluded. The observed number of patients with C&S results in their files could not be used to generate a

representative proportion of de-escalation based on test results. Unknown data regarding some variables such as undergoing surgery, type of surgery, and the theatre could have affected their outcome as possible associated factors for SSTI and AMR. The private wing of the facility though on a small scale involves some specialized medical service units such as gynecology, and pediatrics, among others. However, no disaggregated data was readily available to individually analyze cases of their origin. Other conditions such as comorbidities, wound classes, surgical antimicrobial prophylaxis, and surgeons' experience were not assessed due to data shortage. There was no follow-up of patients to ascertain clinical outcomes post-treatment. This is encouraged for inclusion in future studies to provide a full picture and signify the relationship between practice, risk factors, and the final outcome for better management. Nonetheless, appropriate analytical methods were applied cognizant of ethical requirements. The outcomes of this study create a strong baseline to improve diagnostic stewardship and support the establishment of local treatment guidelines for SSTI and similar clinical conditions. This will improve the antimicrobial stewardship practices for better containment of AMR in the Eastern-central region of Uganda and beyond.

## Conclusions

The prevalence of SSTI was high at Jinja RRH, with only a few cases of SSTI undergoing culture and sensitivity testing. The Gram-negative bacteria were responsible for most of the SSTI while the most isolated pathogen was *Staphylococcus aureus*. Almost half of the infections were due to MDR pathogens including MRSA, possible ESBL-producers, and organisms that are non-susceptible to carbapenems. Given the high resistance observed, laboratory-based diagnosis should be increased so as to use the most appropriate treatment. Infection Prevention and Control strategies also need to be heightened to reduce the prevalence of SSTI. Recognizing SSTI under the GLASS would lead to enhanced surveillance, better preparedness, and response to AMR.

## Acknowledgments

The author appreciates Ms. Zainab Kirunda Kitimbo; Ms. Fauzia Namora; Ms. Sarah Kyoyagala and Ms. Hadia Mukyala for the support provided and reading for clarity.

## Author Contributions

**Conceptualization:** Fahad Lwigale.

**Data curation:** Fahad Lwigale.

**Formal analysis:** Fahad Lwigale, Collins Atuheire, Joseph Kungu, John Bosco Kalule, Immaculate Nabukenya.

**Investigation:** Daniel Kibombo, Morgan Otita, Francis Kakooza.

**Methodology:** Fahad Lwigale, Collins Atuheire.

**Validation:** Fahad Lwigale, Joseph Kungu, John Bosco Kalule, Morgan Otita, Francis Kakooza, Immaculate Nabukenya, Jonathan Mayito, Innocent B. Rwego.

**Visualization:** Simon Dembe Kasango, Jonathan Mayito, Innocent B. Rwego.

**Writing – original draft:** Fahad Lwigale.

**Writing – review & editing:** Fahad Lwigale, Daniel Kibombo, Simon Dembe Kasango, Dickson Tabajjwa, Collins Atuheire, Joseph Kungu, John Bosco Kalule, Morgan Otita, Francis Kakooza, Immaculate Nabukenya, Jonathan Mayito, Innocent B. Rwego.

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
