## [Decision Letter · Decision Letter 0]

2 Apr 2024

PGPH-D-24-00292

Prevalence, resistance profiles and factors associated with skin and soft-tissue infections at Jinja regional referral hospital: A retrospective Study

Dear Dr. Lwigale,

Thank you for submitting your manuscript to PLOS Global Public Health. After careful consideration, we feel that it has merit but does not fully meet PLOS Global Public Health’s publication criteria as it currently stands. Therefore, we invite you to submit a revised version of the manuscript that addresses the points raised during the review process.

We look forward to receiving your revised manuscript.

Kind regards,

Lydia Mosi, Ph.D

Academic Editor

Journal Requirements:

1. Please provide additional details regarding participant consent. If you are reporting a retrospective study of medical records or archived samples, please ensure that you have discussed whether all data were fully anonymized before you accessed them and/or whether the IRB or ethics committee waived the requirement for informed consent. If patients provided informed written consent to have data from their medical records used in research, please include this information.

Additional Editor Comments (if provided):

Reviewers' comments:

Reviewer's Responses to Questions

**Comments to the Author**

1. Does this manuscript meet PLOS Global Public Health’s publication criteria? Is the manuscript technically sound, and do the data support the conclusions? The manuscript must describe methodologically and ethically rigorous research with conclusions that are appropriately drawn based on the data presented.

Reviewer #1: Partly

Reviewer #2: Yes

Reviewer #3: Yes

2. Has the statistical analysis been performed appropriately and rigorously?

Reviewer #1: Yes

Reviewer #2: Yes

Reviewer #3: Yes

3. Have the authors made all data underlying the findings in their manuscript fully available (please refer to the Data Availability Statement at the start of the manuscript PDF file)?

Reviewer #1: Yes

Reviewer #2: Yes

Reviewer #3: Yes

4. Is the manuscript presented in an intelligible fashion and written in standard English?

Reviewer #1: Yes

Reviewer #2: Yes

Reviewer #3: Yes

5. Review Comments to the Author

Reviewer #1: Thank you for inviting me to review this manuscript. I commend the authors for minor work done to publish this interesting research.

1. Abstract: It should be structured

2. Methods – is retrospective study included all variable in the registration that the investigators objective answered?

- What do you if registration chart is incomplete?

- Is the registration book or chart standard for all required information?

3. Analysis methods: - what is your analysis model?

- Model fitness

- VIF( variable inflation factors)

4. Results : what is your response rate?

5. Conclusion : is your conclusion rigorous or strong based your finding?

Reviewer #2: The manuscript titled: “Prevalence, resistance profiles and factors associated with skin and soft-tissue infections at Jinja regional referral hospital: A retrospective Study” addresses an important topic related to the challenges associated with AMR in the healthcare space. The authors find a considerable degree of AMR among patients with wound infections. The authors examine MDR and its associated factors and present a case for a specific set of antimicrobials for the treatment of wound infections. The authors should be applauded for the job well done in general.

I have just a few comments: -

1. Line 48-50 reads “Using microbiology services such as culture and sensitivity (C&S) testing enables determination of the identity of the causative agents and the appropriate antimicrobial agents”. This can be made clearer by cutting down unnecessary words eg “determination or the causative agent…”

2. Line 52: “However, this this is still far from the practice in this setting as most clinical management is still” remove the extra word “this”

3. Line 128-130: reads “The sample size was calculated using the formula (n =p(1-p)z2/d2) and had a power of 80% to estimate the prevalence and factors associated with SSTI at the 95% confidence level.”

4. We need more information on the `prevalence’ and `d’ to be able to replicate the sample size.

5. Line 183 - 186 Bivariate and Multivariate terminologies are used. Please change them to Univariable and multivariable analyses.

6. I would combine table 1 and table 2. There is no need to have them separates as they are. The rows should have the various social demographic and clinical factors while the column should have total, positive and percentage.

7. Some sentences need rewriting eg “In 2019, there were 8.1% (104/1278) cases that underwent laboratory testing for SSTI. Meanwhile, years 2020 and 2021 respectively had 21.1% (226/1072) and 14.3% (196/1370) cases diagnosed by the laboratory to guide management (Fig 2).”

8. The figure and table legends need more information to guide the reader.

9. Line 261: “d (aPR = 1.78, 95%, CI:1.003-3.18, p = 0.04).” ensure consistency in decimal places.

10. Table 5: change bivariate and multivariate to univariable and multivariable

11. The authors found that age was associated with MDR, however, besides contrasting /comparing with previous studies from elsewhere, there is need to provide a substantive explanation/hypothesis on why that is the case. I would expect that children may be exposed to health facilities more and therefore more likely to demonstrate resistance (MDR) compared adults. However, this does not seem to be the case. Did you assess length of hospital stay, HIV status (as a proxy for immunity), wound severity, History of hospital admissions etc. These variables would likely confound age.

Reviewer #3: The scientific report was stressed in the paper, which was also presented. It will be approved after a incorporatent change if the author takes into account all of the comments made in the documents. The feedback provided by the reviewers highlighted areas for improvement, particularly in terms of data analysis and interpretation. Addressing these suggestions will greatly enhance the overall quality and impact of the research findings. Additionally, the reviewers suggested expanding the discussion section to provide more context and relevance to the results. Incorporating this feedback will strengthen the paper and make it more compelling for readers in the scientific community.

6. PLOS authors have the option to publish the peer review history of their article (what does this mean?). If published, this will include your full peer review and any attached files.

**Do you want your identity to be public for this peer review?** For information about this choice, including consent withdrawal, please see our Privacy Policy.

Reviewer #1: No

Reviewer #2: No

Reviewer #3: **Yes: **Desalegn Amneu, Wollega university, Ethiopia

---

## [Editor Report · Decision Letter 1]

18 Jul 2024

Prevalence, resistance profiles and factors associated with skin and soft-tissue infections at Jinja regional referral hospital: A retrospective Study

PGPH-D-24-00292R1

Dear Mr. Lwigale,

We are pleased to inform you that your manuscript 'Prevalence, resistance profiles and factors associated with skin and soft-tissue infections at Jinja regional referral hospital: A retrospective Study' has been provisionally accepted for publication in PLOS Global Public Health.

Best regards,

Lydia Mosi, Ph.D

Academic Editor